# Twelve-Month CPAP Therapy Modulates BDNF Levels in Patients with Severe Obstructive Sleep Apnea: Implications for Metabolic and Treatment Compliance

**DOI:** 10.3390/ijms26125855

**Published:** 2025-06-18

**Authors:** Urszula Karwowska, Aleksandra Kudrycka, Karol Pierzchała, Robert Stawski, Hanna Jerczyńska, Piotr Białasiewicz, Wojciech Kuczyński

**Affiliations:** 1Department of Sleep Medicine and Metabolic Disorders, Medical University of Lodz, Mazowiecka 6/8, 92-215 Lodz, Poland; urszula.karwowska@stud.umed.lodz.pl (U.K.); aleksandra.kudrycka@umed.lodz.pl (A.K.); karol.pierzchala@stud.umed.lodz.pl (K.P.); piotr.bialasiewicz@umed.lodz.pl (P.B.); 2Department of Clinical Physiology, Medical University of Lodz, 92-215 Lodz, Poland; robert.stawski@umed.lodz.pl; 3Central Scientific Laboratory (CoreLab), Medical University of Lodz, 92-215 Lodz, Poland; hanna.jerczynska@umed.lodz.pl

**Keywords:** obstructive sleep apnea, CPAP therapy, brain-derived neurotrophic factor, glucose metabolism, insulin resistance, neurocognitive impairment

## Abstract

Brain-derived neurotrophic factor (BDNF) is a neurotrophin involved in the regulation of synaptic plasticity and metabolic processes, including glucose metabolism and insulin sensitivity. In patients with obstructive sleep apnea (OSA), recurrent episodes of intermittent hypoxia may stimulate BDNF expression as a compensatory neuroprotective response. OSA is associated with metabolic disturbances, such as increased insulin resistance and a higher risk of type 2 diabetes. Continuous positive airway pressure (CPAP) therapy may influence both BDNF levels and metabolic outcomes. The aim of this study was to evaluate changes in BDNF concentration and glucose metabolism in patients with OSA, with particular emphasis on the effect of long-term CPAP therapy. Sixty-six adult patients with OSA confirmed by polysomnography were enrolled and divided into severe (s-OSA) and non-severe (ns-OSA) groups. Fasting blood samples were collected to measure glucose, insulin, and BDNF concentrations. Patients with s-OSA were re-evaluated after 12 months of CPAP therapy and further classified as compliant (sc-OSA) or non-compliant (snc-OSA) based on recorded device usage. The same biochemical parameters were assessed after the 12-month follow-up. Baseline BDNF levels were significantly higher in the s-OSA group compared to the ns-OSA group (20.1 ng/mL vs. 8.1 ng/mL, *p* = 0.02) and correlated with the apnea–hypopnea index (AHI, r = 0.38, *p* = 0.02). In the nsc-OSA group, BDNF concentrations increased significantly after 12 months (16.2 ng/mL vs. 35.5 ng/mL, *p* < 0.001), while no significant change was observed in the sc-OSA group (24.4 ng/mL vs. 27.4 ng/mL, *p* = 0.33). Among sc-OSA patients, a significant improvement in insulin resistance was noted, although no significant changes were observed in fasting glucose or insulin levels. Increased BDNF levels were observed in patients with s-OSA compared to ns-OSA. Compliant CPAP therapy was associated with reduced insulin resistance and no further BDNF increase, in contrast to non-compliance, suggesting a beneficial effect of CPAP on glucose metabolism and BDNF regulation. These findings support the hypothesis that both neurotrophic and metabolic responses in OSA may be modulated by disease severity and therapy adherence.

## 1. Introduction

Brain-derived neurotrophic factor (BDNF) is a crucial member of the neurotrophin family, alongside nerve growth factor, neurotrophin-3, and neurotrophin-4/5, and has emerged as a key modulator of metabolic processes, with conditions like obesity and type 2 diabetes influencing its expression [1,2,3]. Initially, BDNF was shown to enhance brain plasticity [4]. Later studies demonstrated that it also promotes synaptic plasticity, a process critical for learning and memory [5,6,7]. BDNF promotes neurogenesis and supports long-term memory, with reduced levels seen in chronic stress [8,9]. It has been linked to psychiatric disorders such as depression, bipolar disorder, and schizophrenia, potentially influenced by hormonal and metabolic imbalances [1,10]. In Alzheimer’s disease, BDNF may confer neuroprotection by enhancing neuronal survival and reducing neuroinflammation [11,12].

Beyond neurological implications, BDNF influences glucose metabolism in peripheral tissues, enhances insulin sensitivity, and aids in preventing metabolic syndrome [13]. Moreover, metabolic disorders like obesity and type 2 diabetes can alter BDNF expression, which in turn, may contribute to cognitive decline [14].

Given its central role in both neuroplasticity and metabolic regulation, altered BDNF levels have also been implicated in the pathophysiology of obstructive sleep apnea (OSA), which manifests as the repetitive cessation of breathing during sleep due to airway collapse, categorized as hypopneas or apneas. The American Academy of Sleep Medicine defines the apnea–hypopnea index (AHI) threshold for OSA diagnosis, with AHI ≥5 to <15 indicating mild OSA, AHI ≥15 to <30 signifying moderate OSA, and AHI >30 denoting severe OSA. Continuous positive airway pressure (CPAP) stands as the preferred treatment for moderate-to-severe OSA, maintaining upper airway patency. In addition to symptom relief, CPAP may also reduce the risk of dementia [15,16,17]. While CPAP therapy demonstrates efficacy in regulating blood glucose concentration, inconsistencies exist regarding its impact on insulin resistance [18,19]. Treatment success is measured by patient compliance, defined as using CPAP for at least 4 h on at least 70% of nights, with a reduction in AHI of <10 [20]. However, non-adherence to therapy is observed in approximately 30% of patients, with only 45% adherence reported in Polish patients [21].

OSA’s systemic repercussions are substantial, linking it to higher risks of coronary artery disease, myocardial infarction, stroke, hypertension, and atherosclerosis [22,23,24,25]. In addition to these cardiovascular complications, OSA is frequently associated with metabolic disturbances. Studies have shown that greater disease severity correlates with higher fasting glucose levels and increased insulin resistance, as assessed by the Homeostasis Model Assessment of Insulin Resistance (HOMA IR) [18]. HOMA IR is widely accepted in both clinical and research settings for estimating insulin resistance from fasting glucose and insulin levels, providing valuable insights into the metabolic and cognitive consequences often observed in OSA [26,27]. Furthermore, metabolic syndrome is commonly observed in patients with OSA [26]. Psychiatric conditions including depression, anxiety, and mood disorders, as well as cognitive impairments, are also prevalent in this population [17]. Some hypotheses suggest that neurocognitive deficits may be further aggravated by cardiovascular consequences of chronic hyperglycemia [28]. 

Studies examining BDNF in OSA have produced complex and sometimes contradictory findings. For example, refs. [29,30] report that BDNF levels fluctuate according to the severity of nocturnal hypoxia, suggesting that oxygen desaturation may be a key driver of BDNF regulation in OSA. Nevertheless, direct correlations between OSA and BDNF levels remain inconsistent. For example, ref. [31] found significantly elevated BDNF in severe OSA, especially when oxygen desaturation indices were used, while studies relying on the AHI have reported less conclusive results [32,33]. Additional research on CPAP has shown that BDNF levels can undergo rapid changes shortly after treatment initiation. noted a significant early decline in serum and plasma BDNF immediately after the initiation of CPAP, potentially reflecting altered neuronal demand under reduced hypoxic conditions [33], similarly observed a reduction within just one night of using CPAP. However, these changes do not always persist over longer durations, underscoring the dynamic nature of BDNF regulation in OSA.

Our research endeavors to establish a plausible connection between BDNF and glucose concentrations in OSA patients, elucidating intersecting pathways and assessing the impact of long-term CPAP therapy on BDNF alterations.

## 2. Results

The baseline clinical, PSG, and blood parameters from phase 1 are presented in Table 1. The majority of the participants (N = 58, 87.9%) were men. The mean age was 57.1 ± 10.2 years, whereas the mean BMI was 32.4 ± 5.7 kg/m^2^. In s-OSA, there were 23 (88.5%) men, the mean age was 57.4 ± 9.5 years, the mean BMI was 33.4 ± 5.6 kg/m^2^, and the mean neck circumference was 43.3 ± 4.6 cm. In ns-OSA, there were 35 (87.5%) men with a mean age of 56.7 ± 11.3 years, a mean BMI of 30.8 ± 5.5 kg/m^2^, and a mean neck circumference of 42.2 ± 3.6 cm. Although anthropometric variables demonstrated lower values in the non-severe obstructive sleep apnea (ns-OSA) group, no statistically significant differences were observed between the two groups.

Considering PSG results, there was a significant difference between s-OSA and ns-OSA in REM phase length (0.98 ± 0.44 h vs. 1.29 ± 0.55 h, *p* = 0.01), which constituted 17.7 ± 7.3% and 21.4 ± 7.8% of TST, arousal index (51.6 ± 49.4/h vs. 34.7 ± 38.9/h, *p* = 0.01), and AHI (54.1 ± 18.5/h vs. 16.6 ± 7.7/h, *p* < 0.001). The range of AHI in s-OSA was 30.7/h–92.3/h, whereas in ns-OSAS it was 5.0/h–28.6/h. Total sleep time was relatively short in both groups (5.5 ± 1.0 h vs. 5.9 ± 0.9 h, *p* = 0.29). Time of saturation below 90% was longer in the s-OSA group (2.7 ± 1.6 h than in the ns-OSA 1.1 ± 2.0 h, *p* < 0.001), which was equal to 49.4 ± 2.7% and 19.0 ± 36.3% of TST, respectively. Basal saturation was higher in the ns-OSA group (92.3 ± 1.9% vs. 89.1 ± 4.1%, *p* < 0.001).

Regarding blood samples, a significant difference was found in BDNF, but not in glucose and insulin concentrations. Subjects in ns-OSA had a median BDNF concentration of 8.1 (IQR 4.6–17.4) ng/mL, whereas in s-OSA, the median BDNF was higher and equal to 20.1 (IQR 9.0–33.6) ng/mL, *p* = 0.02. This supports the hypothesis that OSA severity is associated with increased BDNF levels, possibly as a compensatory response to intermittent hypoxia. Additionally, a moderate positive correlation was found between BDNF and AHI in the overall population of patients (r = 0.38, *p* = 0.02), as depicted in Figure 1. This correlation suggests that BDNF levels increase in parallel with OSA severity, reinforcing the idea that repeated episodes of hypoxia may stimulate neurotrophic responses.

According to the data stored in the memory of the CPAP devices, among patients in snc-OSA, 8 (50%) patients did not have a single night of CPAP use for longer than 4 h. The other 8 (50%) subjects in snc-OSA exhibited mean CPAP use of at least 4 h for 44.5 ± 20.0% of the nights. In sc-OSA, CPAP was used on average for 88.3 ± 9.6% of the nights. The mean therapeutic pressure was set for 11.4 ± 1.8 cm H_2_O. Data concerning CPAP mask use is presented in Table 2.

For the second phase of the study, anthropometric variables and blood parameters are summarized in Table 3 and Table 4 for sc-OSA and snc-OSA, respectively. No significant change in BDNF was detected in sc-OSA, see Table 3. On the other hand, in the non-compliant group, its median concentration rose from 16.2 (IQR 8.5–29.1) ng/mL to 35.5 (IQR 21.7–39.0) ng/mL, *p* < 0.001. Apart from BDNF, HOMA-IR decreased after a 12-month-long therapy (5.6 ± 4.6 vs. 4.0 ± 1.7, *p* = 0.04) in sc-OSA, see Table 4. However, changes in glucose and insulin concentrations were not significant, and no statistically important correlations were found between BDNF and glucose or insulin levels, as well as HOMA-IR in sc-OSA and snc-OSA.

Comparing sc-OSA and snc-OSA groups after 12 months, no significant differences were found in BMI levels, glucose, insulin, HOMA-IR, and BDNF concentration (results are presented in Table 2).

## 3. Discussion

Frequent and prolonged apneic events lead to intermittent hypoxia, which upregulates BDNF expression in tissues such as the brain and peripheral muscles [30]. The hypoxic environment stimulates the production of BDNF as a compensatory mechanism to protect neurons and promote neuroplasticity, which is particularly crucial in the context of the neurological deficits often observed in OSA patients [27]. This finding is consistent with prior reports indicating that enhanced BDNF expression under hypoxic conditions aids in maintaining synaptic integrity and neuronal survival [31,33].

Our findings align with this, as individuals with severe OSA in our study exhibited significantly higher concentrations of BDNF compared to those with non-severe OSA. Furthermore, we identified a modest correlation between BDNF levels and the apnea–hypopnea index (AHI). Following 12 months of CPAP therapy, BDNF values increased, with a particularly pronounced difference observed in the non-compliant group, where BDNF levels were more than two times higher. This elevated increase in the non-compliant group could be attributed to the limited use of CPAP therapy (less than 4 h per night or not at all), leading to prolonged exposure to hypoxia and greater neuronal damage, which in turn may trigger a compensatory upregulation of BDNF to protect against neurodegeneration. Although a slight increase was also observed in the compliant group, this change did not reach statistical significance.

Interestingly, no statistically significant differences were observed in glucose or insulin levels concerning BDNF concentrations. This lack of association mirrors divergent findings in the literature [19,34] and suggests that the upregulation of BDNF in response to intermittent hypoxia may be independent of immediate changes in glucose metabolism. It should also be acknowledged that the inclusion of a small number of diabetic patients (n = 9) may have introduced confounding. Medications such as metformin or sulfonylureas, commonly used in this group, may independently affect both insulin sensitivity and BDNF expression. Apart from BDNF, HOMA-IR decreased after a 12-month-long therapy, suggesting an improvement in insulin sensitivity among CPAP-adherent patients, which is consistent with prior studies linking CPAP therapy to reductions in systemic inflammation and oxidative stress. CPAP therapy has been shown to reduce insulin resistance and improve metabolic markers, contributing to a lower risk of global cardiovascular disease [35]. This is supported by findings from Baburao and Souza, who observed a tendency towards improved insulin sensitivity after four months of CPAP treatment in severely obese patients [34]. The specific mechanisms through which CPAP therapy affects insulin sensitivity are not fully understood, as existing studies have yielded inconclusive results. However, considering the potential pathways linking sleep apnea to insulin resistance—such as intermittent hypoxia and sympathetic nervous system activation, oxidative stress, and impaired regulation of adipokines production—it can be hypothesized that attenuation of these processes through CPAP therapy may lead to improved tissue sensitivity to insulin [36]. In our study, the compliant group demonstrated a notable reduction in insulin resistance after undergoing CPAP treatment, further reinforcing the role of CPAP in mitigating metabolic complications associated with OSA. Our findings reinforce the potential metabolic benefits of CPAP, the absence of stronger effects on glucose metabolism suggests that additional regulatory mechanisms may be involved, or that a longer follow-up period is needed to observe broader metabolic adaptations. These findings suggest a nuanced relationship between BDNF, OSA severity, and CPAP therapy, underscoring the potential role of BDNF in the pathophysiology of severe OSA and its modulation by CPAP intervention.

Further research is warranted to elucidate the underlying mechanisms and clinical implications of these observations. Studies by [31,33] report marked variability in BDNF responses among OSA patients. This variability is likely due to differences in measurement timing, OSA severity, and patient demographics and underscores the need for standardized methodologies. For example, ref. [31] observed that severe OSA patients with an AHI greater than 30 had significantly higher BDNF levels than healthy controls, with values exceeding those in our cohort. They also found a positive correlation between the Oxygen Desaturation Index (ODI) and BDNF, although this relationship was not examined beyond 90 days of CPAP therapy. Moreover, the limited number of studies addressing BDNF levels in OSA patients undergoing CPAP have yielded inconsistent results. Considering BDNF concentration in the second part of [31], a trend toward a lower value was observed but not significant. However, the ns-OSA group was not controlled. Further research demonstrated no significant difference in BDNF levels between healthy patients and those with newly diagnosed OSA, defined as respiratory disturbance index (RDI) >10/h, >70% obstructive events, and corresponding daytime symptoms [33]. However, after one night of CPAP treatment, BDNF was significantly reduced in patients with severe OSA. Then, after 3 months of therapy, BDNF again increased.

Controls presented with higher concentrations of BDNF than the study group after treatment. Considering studies that did not involve CPAP treatment, we found non-consistent results as well. In the study [32] severe and non-severe OSA patients demonstrated that there was no significant difference between those two groups in BDNF levels. Studies conducted by Panaree [37], as well as Wang [19] reported similar results. In contrast, one Chinese research group showed that OSA patients had lower BDNF levels than the healthy subpopulation [33]. Our findings were consistent with some previous studies but diverged from others, highlighting the inconsistencies in the current literature. Furthermore, analytical platforms differed: our study employed the Luminex xMAP system, while previous investigations primarily relied on ELISA-based methods, which differ in sensitivity and detection range. Collectively, these methodological factors may contribute to the observed variability in BDNF responses despite apparent similarities in clinical parameters.

We studied for the first time a possible link between OSA, glucose metabolism, and BDNF. We also extended the observation time to 12 months, whereas in other studies it was only 3 or 6 months [31,33]. This extended observation period offers valuable insights into the chronic effects of CPAP on BDNF dynamics and metabolic adaptations, which may be overlooked in shorter studies. The absence of statistical significance in certain calculations may indeed be attributed to the relatively modest sample size. While advocating for larger studies is reasonable, it is noteworthy that our study employed a sample size for compliant patients comparable to that of Flores [31] and exceeded that of Staats [33], with an overall higher participant count. Additionally, the intricacy of BDNF concentration dynamics must be acknowledged. Our study included participants presenting with different comorbidities and varying CPAP adherence levels. However, the cohort was predominantly male, and data on educational background and ethnicity were not collected. Future research should aim for a more diverse sample to determine if gender, socioeconomic factors, or ethnicity influence BDNF dynamics and OSA outcomes. Additionally, we recognize that without accurate information on how long participants had suffered from OSA, it is difficult to attribute changes in BDNF specifically to acute versus chronic hypoxia. Future prospective studies should collect detailed OSA history, ideally including symptom onset and prior treatment duration and consider employing biomarkers of chronic hypoxia (e.g., HIF-1α) to clarify the temporal relationship between OSA progression and neurotrophic responses. Precise determination of OSA duration could provide further clarity on how chronic hypoxia affects BDNF levels over time. BDNF levels can respond to many physiological and pathological stimuli. Its increased level in people with OSA may be a compensatory reaction to chronic periods of hypoxia, interrupted by phases of normoxia. CPAP therapy can reduce the hypoxic load, but it does not necessarily lead to the immediate reversal of neuroplastic changes, especially if they are chronic. Although the measurement method we used (Luminex xMAP) is validated and standardized, serum BDNF levels are known to have high diurnal variability and response to external factors.

These multifaceted parameters can contribute to variability in BDNF concentrations. It is imperative to underscore that compliance with CPAP treatment in our study was suboptimal, despite robust recommendations and evident improvements in PSG parameters, aligning with the Polish average. The investigation into the factors influencing compliance rates warrants further exploration to enhance our understanding of the complexities surrounding CPAP adherence and its implications for treatment outcomes. Given that CPAP adherence appears to directly influence both neurotrophic and metabolic responses, future studies should focus on identifying barriers to compliance and developing tailored interventions to optimize treatment outcomes in OSA patients.

## 4. Methods and Materials

### 4.1. Study Population

We enrolled 66 subjects diagnosed with OSA. Inclusion criteria comprised OSA symptoms in the patient’s medical history, along with those reported at the time of admission. Exclusion criteria included a total sleep time of less than 4 h, the presence of central sleep apnea, and severe neurological or cardiovascular diseases. The participants were stratified into two groups: those with severe OSA (AHI > 30, denoted as ‘s-OSA’) and those with non-severe OSA (≥5 AHI <30, denoted as ‘ns-OSA’). CPAP was prescribed for patients in the s-OSA group. Only medications for comorbid conditions were documented in the medical records. For patients with moderate and severe OSA, there were no other interventions aside from prescribed CPAP therapy. Following one year of CPAP therapy, we further categorized the s-OSA group into ‘compliant’ and ‘non-compliant’ subgroups, referred to as ‘sc-OSA’ and ‘snc-OSA’, respectively. This classification was based on the data stored in the CPAP memory, with ‘compliant’ status assigned if the CPAP device was used for an average of at least 4 h per night on at least 70% of the nights during the study period with a reduction in AHI < 10. Conversely, the ‘non-compliant’ status was designated if the CPAP device was used for a shorter duration or AHI ≥ 10. Two patients reported that excessive drying of mucous membranes was the primary reason for their non-compliance with CPAP therapy, while other patients mentioned personal issues. Notably, patients from the ns-OSA group did not participate in the second phase of this study. Alcohol consumption data were not collected in this research. However, smoking history was recorded, with a significant smoking history defined as at least 15 pack–years. Among the patients with an AHI > 30, 18 (45%) were identified as smokers, of which 11 demonstrated good compliance with CPAP therapy, while 7 showed poor compliance. In the group with AHI < 30, 7 patients (26.9%) were identified as smokers. Based on the medical histories, 9 patients (13.6%) received a diagnosis of type 2 diabetes mellitus (DM2). Within the s-OSA group, 7 patients (17.5%) had DM2, whereas in the ns-OSA group, there were 2 patients (7.7%) with DM2. All individuals with DM2 were undergoing treatment with oral antidiabetic drugs, and none were utilizing insulin injections. Specifically, in the s-OSA group, 3 patients were prescribed sulfonylurea, and 4 were on metformin. In the ns-OSA group, all subjects were treated with metformin. In the second phase of the study, 6 patients with DM2 were categorized as sc-OSA, while only 1 patient as a snc-OSA.

### 4.2. Data Collection

The study was conducted in the Department of Sleep Medicine and Metabolic Disorders in Medical University in Lodz, Poland. The data was collected between November 2019 and March 2021. All subjects were informed about the purpose of the study and signed the informed consent. The study was approved by the Ethics Committee of Medical University of Lodz (RNN/23/15/KE and RNN/47/14/KE; RNN/92/15/KE). The study was funded by an institutional grant from the Medical University of Lodz 503/0-079-06/503-01.

### 4.3. Study Protocol

The study comprised two distinct phases, as illustrated in Figure 2. In the initial phase, each participant underwent a comprehensive examination to collect anthropometric data, including sex, weight, height, and neck circumference. Polysomnography was performed using the Nox A1s™ PSG System 2022 (Nox Medical, Reykjavík, Iceland), a wireless PSG device designed for both laboratory and home sleep studies. The system recorded electroencephalography (EEG) signals using electrodes placed at O1, O2, C3, C4, Fp1, Fp2, A1, and A2, along with electrocardiography (ECG), electromyography (EMG), electrooculography (EOG), respiratory effort using Nox RIP belts on the chest and abdomen, airflow via a nasal cannula, and oxygen saturation (SpO_2_) measured with a Nox oximeter. Throughout the medical histories, patients provided information regarding comorbidities, with a specific focus on DM 2, neurodegenerative disorders, and cardiovascular diseases. All participants initiated sleep at approximately 10 pm, and in the subsequent morning, at around 6 am, a fasting venous blood sample was procured to assess concentrations of glucose, insulin, and BDNF. Serum samples were prepared by centrifugation at 3200× *g* for 10 min at 4 °C and stored at −80 °C until assaying. Additionally, insulin resistance was evaluated using the HOMA IR.

Serum BDNF levels were measured using a Luminex xMAP technology-based kit (cat. no. HMYOMAG-56K, Millipore, Burlington, MA, USA) according to the manufacturer’s instructions. Standards, samples, and BDNF antibody-coated magnetic beads were added to the wells. During overnight incubation, the analyte from the samples was captured by the antibodies. After a washing step, a biotinylated detection antibody was added. After incubation, unbound antibodies were washed off, and a streptavidin–PE conjugate was added to detect biotinylated antibodies on the surface of each bead. After an additional incubation step, any unbound conjugates and free dyes were removed by washing, and the beads were resuspended in the Drive Fluid buffer. Two types of fluorescence were measured using a Luminex^®^ MAGPIX^®^ analyzer (Luminex, Austin, TX, USA): one to identify the analyte and the other to determine the intensity of the PE signal, which is directly proportional to the amount of bound analyte. All results were analyzed using Belysa 1.1.0 software, and protein concentrations (pg/mL) were determined by interpolation from the standard curve based on a five-parameter logistic model (5-PL). All measurements were performed in duplicate. The lower limit of detection (LOD) for BDNF was 0.2 pg/mL, and the study showed high sensitivity and repeatability. The mean intra-assay coefficients of variation (CVs) for BDNF were 4.5%, indicating low analytical variability and high measurement precision.

PSG results underwent manual signal analysis conducted using Noxturnal™ Software (Nox Medical ver 7.0, Reykjavík, Iceland), which provided sleep staging, respiratory event detection, and signal quality control. PSG recordings were manually scored by a trained sleep technician following AASM 2020 guidelines to determine OSA severity, sleep architecture, and respiratory disturbance indices. Hypopnea events were defined according to AASM 2020 criteria as a ≥30% reduction in airflow lasting at least 10 s, associated with either a ≥3% oxygen desaturation or an arousal. Treatment plans were individually tailored, with mandatory prescription of CPAP for individuals exhibiting an AHI exceeding 30 and treatment was initiated using an auto-adjusting positive airway pressure (autoCPAP) device. The second phase of the study was carried out twelve months later, exclusively involving participants from the severe OSA (s-OSA) group. A medical professional conducted a thorough examination, measuring anthropometric parameters. Subsequently, a fasting venous blood sample was obtained to replicate the assessments of glucose, insulin, and BDNF concentrations.

### 4.4. Statistical Analysis

Statistical analyses were performed with Statistica 13.3 (Statsoft, Cracow, Poland). *t*-test was used for Gaussian distribution, whereas the U-Mann–Whitney test was for non-normal distribution. Data distribution was checked with the Shapiro–Wilk test. The Chi^2^ test was used for categorical data, which were then presented as a frequency (percent). For dependent pairs, Wilcoxon matched pairs test was used. The correlation was calculated with Pearson’s index.

## 5. Conclusions

Despite the importance of our findings, several limitations warrant consideration. First, while our sample size was comparable to previous studies, a larger and more diverse population would enhance the robustness of our conclusions. Second, although diabetes medications were recorded, we did not account for other potential confounders such as diet, physical activity, stress levels, and additional pharmacological treatments that may have influenced BDNF levels. Third, CPAP compliance was suboptimal, and the reasons for nonadherence were self-reported, introducing the possibility of bias. Moreover, although this study spanned 12 months, extending the follow up could reveal more insights into long-term BDNF trends. Finally, although we focused on BDNF as a key biomarker, future research should explore additional markers related to metabolic and neurocognitive outcomes in OSA patients. Taken together, our findings highlight a complex interplay between OSA severity, CPAP compliance, and BDNF regulation, with implications for both metabolic and neurocognitive outcomes.

## Figures and Tables

**Figure 1 ijms-26-05855-f001:**
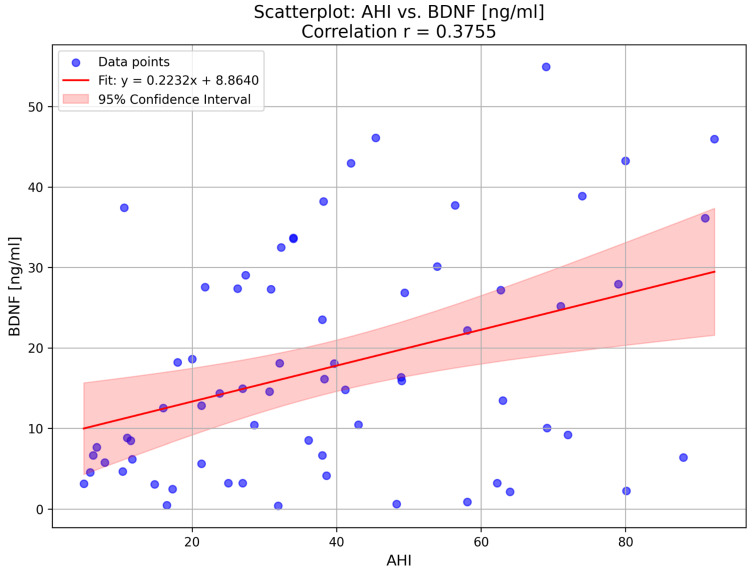
Correlation between BDNF (ng/mL) and AHI (events/hour) in the overall group. A moderate positive correlation was observed (r = 0.38, *p* < 0.05). The regression equation is: BDNF (ng/mL) = 8.86 + 0.22 × AHI. The red line represents the linear regression model, and the shaded area shows the 95% confidence interval.

**Figure 2 ijms-26-05855-f002:**
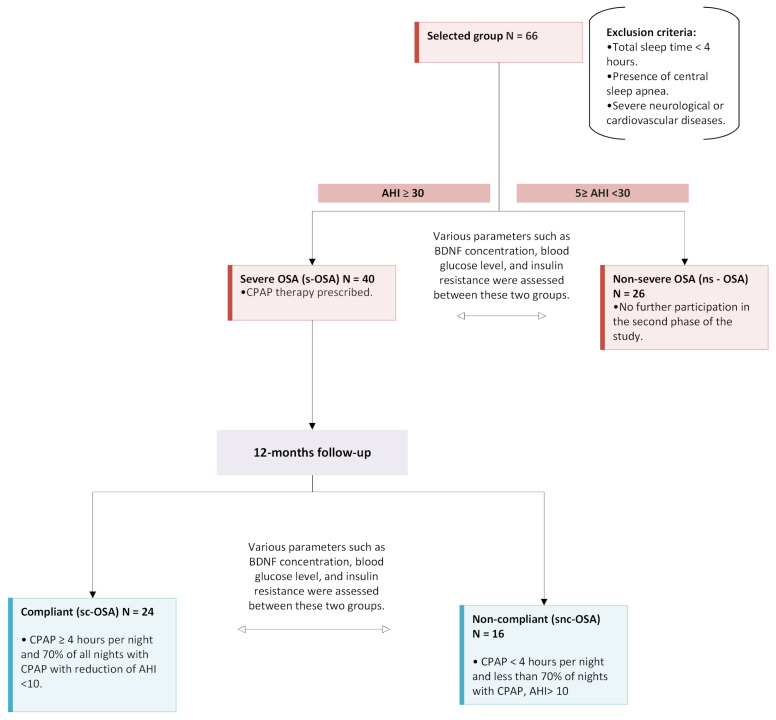
The flowchart illustrates the classification of 66 patients with OSA. Patients were divided based on AHI into severe (s-OSA) and non-severe (ns-OSA) groups. After one year of CPAP therapy, s-OSA patients were further classified into compliant (sc-OSA) and non-compliant (snc-OSA) based on CPAP usage. Various parameters like BDNF concentration and glucose metabolism were assessed between groups.

**Table 1 ijms-26-05855-t001:** All parameters, except for BDNF are presented as a mean ± standard deviation unless otherwise indicated. BDNF is presented as the median (IQR). *p* values were calculated for s-OSA and ns-OSA. Abbreviations: BMI—body mass index; TST—total sleep time; REM—rapid eye movement sleep; NREM—non-rapid eye movement sleep; AHI—apnea–hypopnea index; SpO_2_—saturation; HOMA-IR—homeostatic model assessment—insulin resistance.

	Overall	s-OSA	ns-OSA	*p*
N (%)	66	40 (60.6%)	26 (39.4%)	
Male sex, *n* (%)	58 (87.9%)	35 (87.5%)	23 (88.5%)	0.61
Age (years)	57.1 ± 10.2	57.4 ± 9.5	56.7 ± 11.3	0.79
BMI (kg/m^2^)	32.4 ± 5.7	33.4 ± 5.6	30.8 ± 5.5	0.1
Neck circumference (cm)	42.9 ± 4.2	43.3 ± 4.6	42.2 ± 3.6	0.07
TST (hours)	5.66 ± 0.95	5.53 ± 0.99	5.86 ± 0.88	0.29
REM (hours)	1.10 ± 0.51	0.98 ± 0.44	1.29 ± 0.55	0.01
REM %TST	19.4 ± 7.7%	17.7 ± 7.3%	21.4 ± 7.8%	0.00
NREM (hours)	4.56 ± 0.8	4.55 ± 0.92	4.57 ± 0.6	0.93
NREM %TST	80.9 ± 7.7%	82.4 ± 7.3%	78.6 ± 7.8%	0.14
Arousal index (per hour)	44.9 ± 46.0	51.6 ± 49.4	34.6 ± 38.9	0.00
AHI (per hour)	39.7 ± 23.8	54.1 ± 18.5	16.6 ± 7.7	0.00
Basal SpO_2_ (%)	90.3 ± 3.7	89.1 ± 4.1	92.3 ± 1.9	0.00
Time below 90% SpO_2_ (min)	126.5 ± 115.1	163.9 ± 96.5	66.8 ± 119.1	0.00
Time below 90% SpO_2_ (%TST)	37.4 ± 34.3%	49.4 ± 2.7%	19.0 ± 36.3%	0.00
Glucose (mg/dL)	110.5 ± 21.2	113.9 ± 4.8	105.4 ± 13.0	0.21
Insulin (mIU/L)	15.7 ± 11.1	17.6 ± 13.5	12.9 ± 5.1	0.39
HOMA-IR	4.44 ± 3.63	5.12 ± 4.40	3.39 ± 1.50	0.32
BDNF (ng/mL)	14.7 (5.9–27.8)	20.1 (9.0–33.6)	8.1 (4.6–17.4)	0.02
Diabetes mellitus t2, *n* (%)	9 (13.6%)	7 (17.5%)	2 (7.7%)	0.44

**Table 2 ijms-26-05855-t002:** All parameters, except for BDNF, are presented as a mean ± standard deviation. BDNF is presented as median (interquartile range, IQR). Data represent outcomes after 12 months among (sc-OSA) and (snc-OSA). Abbreviations: HOMA-IR—homeostatic model assessment—insulin resistance; CPAP—continuous positive airway pressure; AHI—apnea–hypopnea index.

	sc-OSAS	snc-OSAS	*p*
BMI (kg/m^2^)	33.7 ± 5.7	30.0 ± 9.3	0.30
Glucose (mg/dL)	111.4 ± 21.6	106.4 ± 13.6	0.69
Insulin (mIU/L)	14.6 ± 5.7	15.2 ± 7.3	0.92
HOMA-IR	4.0 ± 1.7	4.0 ± 1.9	0.92
BDNF (ng/mL)	27.4 (14.8–33.8)	35.5 (21.7–39.0)	0.13
Therapeutic pressure in CPAP (mmH_2_O)	11.4	11.2	0.00
AHI (per hour)	8.6 ± 10.7	12.5 ± 18.1	0.68
Nights with CPAP (%)	88.3 ± 9.6%	44.5 ± 20.0%	0.00

**Table 3 ijms-26-05855-t003:** All parameters, except for BDNF, are presented as a mean ± standard deviation. BDNF is presented as median (interquartile range, IQR). Data apply to before CPAP and after 12 months of CPAP. Abbreviations: HOMA-IR—homeostatic model assessment—insulin resistance; CPAP—continuous positive airway pressure; AHI—apnea–hypopnea index.

	sc-OSAS (Compliant) n = 24		*p*
	Before CPAP	After 12 Months of CPAP	
BMI (kg/m^2^)	34.1 ± 6.2	33.7 ± 5.7	0.57
Glucose (mg/dL)	117.3 ± 28.9	111.4 ± 21.6	0.08
Insulin (mIU/L)	18.1 ± 11.3	14.6 ± 5.7	0.09
HOMA-IR	5.6 ± 4.6	4.0 ± 1.7	0.04
BDNF (ng/mL)	24.4 (9.7–34.3)	27.4 (14.8–33.8)	0.33

**Table 4 ijms-26-05855-t004:** All parameters, except for BDNF, are presented as a mean ± standard deviation. BDNF is presented as median (interquartile range, IQR). Abbreviations: HOMA-IR—homeostatic model assessment—insulin resistance; CPAP—continuous positive airway pressure; AHI—apnea–hypopnea Index.

	snc-OSAS (Non-Compliant), n = 16		*p*
	Before CPAP	After 12 Months of CPAP	
BMI (kg/m^2^)	32.4 ± 4.6	30.0 ± 9.3	0.36
Glucose (mg/dL)	108.8 ± 16.4	106.4 ± 13.6	0.33
Insulin (mIU/L)	16.9 ± 16.7	15.2 ± 7.3	0.88
HOMA-IR	4.4 ± 4.1	4.0 ± 1.9	0.88
BDNF (ng/mL)	16.2 (8.5–29.1)	35.5 (21.7–39.0)	0.00

## Data Availability

The data presented in this study are available on request from the corresponding author due to (specify the reason for the restriction).

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
