# Peer review of "Twelve-Month CPAP Therapy Modulates BDNF Levels in Patients with Severe Obstructive Sleep Apnea: Implications for Metabolic and Treatment Compliance"

_ijms, 2025, doi:10.3390/ijms26125855_

Round 1
Reviewer 1 Report
Comments and Suggestions for Authors
Obstructive sleep apnea (OSA) is a common disorder characterized by repeated episodes of upper airway collapse during sleep, leading to intermittent hypoxia. This hypoxia can result in neuronal damage, increased risk of cognitive impairment, chronic inflammation, and metabolic dysregulation. Previous studies have shown that brain-derived neurotrophic factor (BDNF) is stimulated by intermittent hypoxia and may provide neuroprotection under low oxygen conditions.
The study titled “Twelve-Month CPAP Therapy Modulates BDNF Levels in Patients with Severe Obstructive Sleep Apnea: Implications for Metabolic and Cognitive Outcomes” investigates the relationship between OSA severity (measured by AHI) and BDNF levels, and how these may change following long-term treatment. The authors report a positive, albeit modest, correlation between AHI and circulating BDNF levels, consistent with previous findings. However, they observe that while 12 months of CPAP treatment significantly improved AHI, it did not result in a reduction in BDNF concentrations.
Overall, this is a clearly structured and valuable study. The 12-month follow-up provides important insights into how BDNF dynamics may evolve under chronic OSA conditions, potentially offering biomarkers for clinical diagnosis and therapeutic targets. However, several issues need to be addressed before publication:
-
Hypoxia validation: To better support the role of BDNF as a response to hypoxia, a biomarker such as HIF-1α should be included in the study. This would help confirm whether the patients were indeed experiencing intermittent hypoxia due to OSA.
-
Clarification of BDNF trends: The data in Tables 2–4 are somewhat unclear. If BDNF is upregulated by intermittent hypoxia, one would expect its levels to decrease following CPAP treatment due to reduced hypoxic events. Please clarify this paradox in the discussion. Additionally, a non-CPAP control group would provide important comparative data on BDNF trends.
-
Mechanism of metabolic improvement: There is insufficient discussion regarding the mechanisms by which CPAP therapy improves insulin resistance. Is this improvement attributed primarily to better sleep quality, reduced inflammation, or the alleviation of hypoxia? This should be addressed more explicitly in the discussion.
-
Minor concerns: Several formatting and typographical errors are present in the text. A thorough proofreading is recommended prior to resubmission.
English in this manuscript could be improved. For example, some of the sentences are wordy.
Author Response
- Hypoxia validation: To better support the role of BDNF as a response to hypoxia, a biomarker such as HIF-1α should be included in the study. This would help confirm whether the patients were indeed experiencing intermittent hypoxia due to OSA.
Comment : We agree that including such markers would strengthen the evidence for the role of BDNF as a response to intermittent hypoxia in OSA. Unfortunately, we do not have access to data on HIF-1α or other hypoxia biomarkers in our current study cohort. We acknowledged it in the discussion.
Line : 436
- Clarification of BDNF trends: The data in Tables 2–4 are somewhat unclear. If BDNF is upregulated by intermittent hypoxia, one would expect its levels to decrease following CPAP treatment due to reduced hypoxic events. Please clarify this paradox in the discussion. Additionally, a non-CPAP control group would provide important comparative data on BDNF trends.
Comment 1: Although a decrease in BDNF was expected based on the provided research, a statistically non-significant increase in the compliant group was observed (p=0.33). We included this information in the manuscript.
Comment 2: Unfortunately, we do not have data from patients in the suggested group. All patients diagnosed with severe OSA were offered the treatment. According to the completed questionnaires, no patient fully declined the CPAP treatment. Even if a patient did not use the device for more than 4 hours on any given night, there were several attempts at use that lasted for shorter durations.
- Mechanism of metabolic improvement: There is insufficient discussion regarding the mechanisms by which CPAP therapy improves insulin resistance. Is this improvement attributed primarily to better sleep quality, reduced inflammation, or the alleviation of hypoxia? This should be addressed more explicitly in the discussion.
Comment : We revised the manuscript and included the suggested information.
Lines : 357 - 364
- Minor concerns: Several formatting and typographical errors are present in the text. A thorough proofreading is recommended prior to resubmission.
Comment : We reformatted the text and corrected the errors
Reviewer 2 Report
Comments and Suggestions for Authors
This manuscript examines the alterations of BDNF levels and glucose metabolism in OSA patients with CPAP therapy over a period of 12 months. Although the paper offers some perspectives, there are some concerns that affect the outcomes of the study.
1. The finding is less generalizable because most of the participants were male.
2. OSA is often a chronic condition, and without accurate information on its duration, it is challenging to discern whether the observed effects are due to acute or chronic hypoxia, or to disentangle the temporal relationship between OSA progression and BDNF changes.
3.The authors acknowledge that their findings are inconsistent with some prior studies but do not provide a sufficiently in-depth analysis of these discrepancies.
4. The inclusion of 9 diabetic patients introduces confounding. Patients taking antihypertensive drugs such as metformin and sulfonylureas may have independent impacts on insulin resistance and expression of BDNF.
5. It is shown in Table 1 that BDNF is reported as median but group comparisons are made using t-tests which is inappropriate for normally distributed data. Non-parametric tests such as Mann-Whitney U shold be used instead.
6. The title and abstract reference “implications for metabolic and cognitive outcomes,” but the study neglects cognitive assessments.
Author Response
- The finding is less generalizable because most of the participants were male.
Comment : We have already acknowledge this limitation in the manuscript by explicitly noting the male predominance of our cohort and highlighting the need for more diverse samples in future studies.
- OSA is often a chronic condition, and without accurate information on its duration, it is challenging to discern whether the observed effects are due to acute or chronic hypoxia, or to disentangle the temporal relationship between OSA progression and BDNF changes.
Comment : We thank the reviewer for highlighting this important point. We have now explicitly acknowledged this limitation in the Discussion
Lines : 421-427
3.The authors acknowledge that their findings are inconsistent with some prior studies but do not provide a sufficiently in-depth analysis of these discrepancies.
Comment : We thank the reviewer for the comment. The Discussion section has been expanded to include a more detailed comparison with prior studies.
Lines : 400 - 404
- The inclusion of 9 diabetic patients introduces confounding. Patients taking antihypertensive drugs such as metformin and sulfonylureas may have independent impacts on insulin resistance and expression of BDNF.
Comment : Thank you for this observation. We have added a sentence in the Discussion acknowledging the potential confounding effect of diabetes and related medications on insulin sensitivity and BDNF expression.
Lines : 346 - 350
- It is shown in Table 1 that BDNF is reported as median but group comparisons are made using t-tests which is inappropriate for normally distributed data. Non-parametric tests such as Mann-Whitney U shold be used instead.
Comment : Thank you for your observation. We would like to clarify that only BDNF exhibited a non-normal distribution; therefore, we reported it using the median and interquartile range (IQR) and applied a non-parametric test (Mann-Whitney U) for group comparison. For all other variables, the distribution was assessed and found to be normal, which justified the use of parametric tests (t-tests) along with mean and standard deviation.
- The title and abstract reference “implications for metabolic and cognitive outcomes,” but the study neglects cognitive assessments.
Comment : Thank you for your observation. We have revised the title to better reflect the scope and focus of our study.
Line : 4
Round 2
Reviewer 1 Report
Comments and Suggestions for Authors
most of my concerns were addressed by the authors and I have no further questions before its publication.
Reviewer 2 Report
Comments and Suggestions for Authors
The author has adequately addressed the raised questions. Recommended for publication.